# Trajectories of Allopregnanolone and Allopregnanolone to Progesterone Ratio across the Six Subphases of Menstrual Cycle

**DOI:** 10.3390/biom13040652

**Published:** 2023-04-05

**Authors:** Ajna Hamidovic, John Davis, Fatimata Soumare, Avisek Datta, Aamina Naveed

**Affiliations:** 1College of Pharmacy, University of Illinois at Chicago, 833 S. Wood St., Chicago, IL 60612, USA; 2College of Medicine, University of Illinois at Chicago, 1601 W. Taylor St., Chicago, IL 60612, USA; 3School of Public Health, University of Illinois at Chicago, 1603 W. Taylor St., Chicago, IL 60612, USA

**Keywords:** neuroactive steroid hormones, menstrual cycle, allopregnanolone, progesterone, 5α-reductase

## Abstract

*Background*: Allopregnanolone is one of the most studied neuroactive steroids; yet, despite its relevance to neuropsychiatric research, it is not known how it, as well as its ratio to progesterone, varies across all six subphases of the menstrual cycle. Two enzymes—5α-dihydroprogesterone and 5α-reductase—convert progesterone to allopregnanolone, and, based on immunohistochemical studies in rodents, the activity of 5α-reductase is considered the rate-limiting step in the formation of allopregnanolone. It is not clear, however, whether the same phenomenon is observed across to the menstrual cycle, and, if so, at what point this takes place. *Methods*: Thirty-seven women completed the study during which they attended eight clinic visits across one menstrual cycle. We analyzed their allopregnanolone and progesterone serum concentrations using ultraperformance liquid chromatography–tandem mass spectrometry, and we implemented a validated method to realign the data from the original eight clinic study visits, following which we imputed the missing data. Hence, we characterized allopregnanolone concentrations, and the ratio of allopregnanolone:progesterone at six menstrual cycle subphases: (1) early follicular, (2) mid-follicular, (3) periovulatory, (4) early luteal, (5) mid-luteal, and (6) late luteal. *Results*: There were significant differences in allopregnanolone levels between (1) early follicular and early luteal, (2) early follicular and mid-luteal, (3) mid-follicular and mid-luteal, (4) periovulatory and mid-luteal, and (5) mid-luteal and late luteal. We detected a sharp drop in allopregnanolone:progesterone ratio in the early luteal subphase. Within the luteal subphase, the ratio was the lowest in the mid-luteal subphase. *Conclusions*: Allopregnanolone concentrations are the most distinct, relative to the other subphases, in the mid-luteal subphase. The shape of the allopregnanolone trajectory across the cycle is similar to that of progesterone; however, the proportion of the two neuroactive steroid hormones is drastically different due to enzymatic saturation, which takes place at the start of the early luteal subphase, but continuing through, and peaking, in the mid-luteal subphase. Hence, the estimated activity of 5α-reductase decreases, but does not cease, at any point across the menstrual cycle.

## 1. Introduction

3α-hydroxy-5α-pregnan-20-one, or allopregnanolone, is one of the most studied neuroactive steroids. It was identified in 1938 [1], with the most recent major event in 2019 involving the Food and Drug Administration (FDA) approval of Zulresso (brexanolone)—an analog of allopregnanolone—intravenous injection for the treatment of postpartum depression (PPD) in adult women. 

Allopregnanolone is synthesized from progesterone in two steps (Figure 1). First, 5α- and 5β-reductases convert progesterone to 5α-dihydroprogesterone (5α-DHP) and 5β-dihydroprogesterone (5β-DHP), respectively, following which 3α-hydroxysteroid dehydrogenase (3α-HSD) converts 5α-DHP and 5β-DHP to allopregnanolone and pregnanolone, respectively [2,3]. Which pregnanedione (5α- or 5β-) is a substrate for 5α-HSD has not been specifically examined to date, but the process is presumably related to the differential tissue distribution of 5α-DHP or 5β-DHP, which is, in turn, determined by differential expression profiles of the steroidogenic enzymes 5α- and 5β-reductases [4].

The activity of 5α-reductase is considered the rate-limiting step in the conversion from progesterone to allopregnanolone [5,6]. This is based on the results of immunohistochemical studies showing that the allopregnanolone decrease in corticolimbic areas [7,8,9] is related to a decreased expression of 5α-reductase [8,10,11]. Whether, and when, this occurs in the menstrual cycle is not clear, and, importantly, whether the estimated activity of 5α-reductase ceases at any point of the menstrual cycle is not known. The activity of 5α-reductase is estimated in the present study since it is not directly involved in the formation of allopregnanolone from progesterone (Figure 1). 

Throughout decades of research, in human and animal models, allopregnanolone was evaluated for its potent gamma aminobutyric acid A (GABAA) receptor allosteric modulatory effects, and shown to modulate several neuropsychiatric conditions [12,13]. Nonetheless, some of the most fundamental questions related to endogenous allopregnanolone trajectories remain unanswered. For example, it is not known how allopregnanolone, as well as the ratio of allopregnanolone to progesterone, vary across the entire menstrual cycle; that is, the six subphases of the menstrual cycle: (1) early follicular, (2) mid-follicular, (3) periovulatory, (4) early luteal, (5) mid-luteal, and (6) late luteal. Given its potent effects on the GABAA receptor, understanding allopregnanolone variability across the entire menstrual cycle has clinical relevance to conditions such as epilepsy [14] and premenstrual dysphoric disorder [15]. The assessment of changes in the ratio of allopregnanolone to progesterone approximates the activity of 5α-reductase in the conversion from progesterone to allopregnanolone, thereby broadening our understanding of enzymatic mechanisms regulating allopregnanolone production. 

The study of menstrual-cycle trajectories is complicated by the fact that menstrual-cycle duration varies both within and between women [16]. Hence, it is now known when study participants should be scheduled to attend the study visits corresponding to the six menstrual cycle subphases that capture distinct changes in the circulating sex hormone (estradiol and progesterone) levels. An accurate method for dealing with this complexity was developed and validated in the BioCycle study—designed to evaluate associations between endogenous hormones across the menstrual cycle and oxidative stress, inflammation, and metabolic biomarkers in premenopausal women [17]. It involves collecting data from eight clinic visits at the estimated menstrual cycle subphases—early follicular, mid-follicular, periovulatory (three visits to capture the short-lived serum luteinizing hormone (LH) peak), early luteal, mid-luteal, and late luteal. Data are then first realigned *post hoc* based on the periovulatory LH peak, and imputed to account for the missing data resulting from the realignment [17]. The protocol is implemented in lieu of the daily specimen collection, which is impractical and costly. 

We implemented this refined method for staging menstrual cycle subphases, and analyzed allopregnanolone and progesterone using mass spectrometry, which is preferable to immunoassay technology because it more precisely recognizes similar structures [18,19,20], to clarify the changes of allopregnanolone and the enzymatic activity involved in its synthesis across the entire menstrual cycle.

## 2. Materials and Methods

### 2.1. Study Design

A detailed description of the study is published in [21]. In summary, we enrolled reproductive-age women to participate in the Premenstrual Hormonal and Affective State Evaluation (PHASE) project, which contains two sub-studies; attendance at eight menstrual-cycle clinic visits across the menstrual cycle, and participation in the Trier social stress test in the mid-late luteal subphase of the menstrual cycle. The present publication summarizes allopregnanolone and progesterone data from the eight clinic visits. PHASE is a registered clinicaltrials.gov (accessed on 9 February 2023) study (NCT03862469). 

### 2.2. Study Sample

In PHASE, we enrolled women who did not use illicit drugs, did not smoke, and were not heavy alcohol drinkers, who were between the ages of 18 to 35 and had regular cycles lasting between 21 and 35 days [19], and who were not on any prescription medications and did not have a current anxiety or depression disorder assessed by the Structured Clinical Interview for DSM Disorders (SCID). Detailed study inclusion/exclusion criteria are published in [21].

### 2.3. Study Procedures

Study participants used the Clearblue urine tests to self-test urinary luteinizing hormone peak [22]. This self-testing was not for fertility purposes; we implemented this procedure to guide the scheduling of study visits in the last menstrual cycle (last paragraph in this section).

On a daily basis, study participants uploaded the result screenshot of the Clearblue app into REDCap. We monitored the adherence to the daily urine LH testing procedure, which is very important for the capturing of the short-lived LH surge [23]. In the event a participant missed one daily test, the research coordinator contacted her to resolve any potential issues, and to ask the participant to complete the procedure in a timelier manner.

As part of the study protocol [21], study participants attended eight clinic visits [17] (Appendix A), at which, among other measures, we collected serum samples for the purpose of analyzing allopregnanolone and progesterone levels. 

### 2.4. Sample Analysis 

#### 2.4.1. Progesterone and Allopregnanolone

The analyses of progesterone and allopregnanolone were performed by the Mass Spectrometry Core in the Research Resources Center of the University of Illinois at Chicago. The limits of detection of allopregnanolone and progesterone ranged from 5 to 25 pg and 0.5 to 1.5 pg, respectively. The lower limit of quantification of allopregnanolone and progesterone ranged from 25 to 50 pg and 1.5 to 2.5 pg, respectively. Each calibration standard’s accuracy was within the acceptable range of 15%. The recovery of allopregnanolone and progesterone was assessed for quality-control samples at three levels (low, mid, and high) during the initial method development. For allopregnanolone, recovery of 75 pg, 300 pg, and 600 pg was 99.8%, 101.1%, and 105.6%, respectively. For progesterone, recovery of 7.5 pg, 20 pg, and 40 pg was 93.5%, 110.1%, and 95.8%, respectively. Allopregnanolone and allopregnanolone-D4 internal standards were supplied by Steraloids (Newport, RI, USA). Progesterone and Progesterone-D9 internal standards were supplied by Cerilliant (Round Rock, TX, USA). 

#### 2.4.2. Luteinizing Hormone

Luteinizing hormone assays were performed by CLIA-certified ARUP Laboratories (Salt Lake City, UT, USA) using a quantitative electrochemiluminescent immunoassay [24,25]. During the first incubation, 20 μL of specimen was used to form a sandwich complex. The limit of detection of LH is 0.3 mIU/mL and the lower limit of quantification of LH is 1 mIU/mL. In validation studies performed by ARUP Laboratories, mean recovery percentage of LH was 102.0 ± 2.5, intra-assay coefficient of variation values were all 0.5%, and inter-assay coefficient of variation values ranged from 0.9% to 1.5%.

### 2.5. Power and Data Analyses

We conducted a power analysis to test the hypothesis for one independent group mean against the null hypothesis using a two-tailed test, a medium effect size (d = 0.50), and an alpha of 0.05. Results showed that a total sample of 32 participants was required to achieve a power of 0.80.

We proceeded with completing all study analyses using the R software [26]. To ensure that study participants were aligned to the same biological window, we first realigned the visit data as outlined in the Biocycle study [17]. We considered LH surge to have occurred when serum LH value reached 14.0 IU/L or greater [24]. We then log-transformed the data, following which we implemented the multiple imputation methodology [27] to deal with the missing data. We created 10 datasets using the mice package in R [28], and conditioned on age, race, body mass index, and adjacent hormone measurements [17]. We conducted pairwise comparisons of all timepoints for both the realigned and imputed data, and we used the false discovery rate (FDR) to correct for multiple comparisons. 

## 3. Results

### 3.1. Study Sample, Realignment, and Imputation Results 

Allopregnanolone concentrations were analyzed for two participants in a separate batch. Those values were returned as being on average between 5 and 10 times (depending on the timepoint) the values of the remaining study participants analyzed in batches separate from that run. Hence, the data from those two participants was considered a lab error and excluded from the analysis. The analyzable dataset, hence, constituted 35 participants, out of the 37 participants who completed the study.

Of the remaining 35 participants, 5 had anovulatory cycles confirmed by the absence of a progesterone value ≥ 5 ng/mL [17]. As our analytical approach ensured a confirmed ovulatory cycle, the analysis was, therefore, completed on 30 study participants. Mid-cycle (periovulatory) LH levels (>14.0 IU/L) [24] were captured in 23 study participants. The serum LH mean and standard deviation levels for the 23 participants with detectable peak LH were 34.24 and 14.88, respectively. For those 23 participants, the data were realigned as in the Biocycle study (Appendix A). The peaks were captured at one of the three periovulatory phase visits (visits 3, 4, or 5) for 18 individuals. Of the remaining participants, 4 participants’ LH peak was captured on visit 6, and 1 participant’s LH peak was captured on visit 7. We realigned the visit data for the 7 participants who did ovulate (based on their luteal progesterone values), but whose LH peak was missed (i.e., undetectable on any of the blood draw visits) on an individual basis, as specified in Mumford et al. [17], and further described in Hamidovic et al. [21].

### 3.2. Study Participants 

The present analysis included 30 participants with ovulatory cycles (Table 1), approximately 26 years old with 40% of the sample consisting of White participants. The participants were on average approximately between the normal and overweight BMI status, with the average age of menarche being approximately 12 years old. We present further details in Table 1. 

### 3.3. Allopregnanolone and Progesterone 

We present the trajectory of allopregnanolone, progesterone, and the ratio as mean and variance at each of the six menstrual cycle subphases in Figure 2a–c, respectively. Results of FDR-corrected pairwise comparisons of the imputed dataset across all the timepoints showed that there were significant differences in allopregnanolone levels between (1) early follicular and early luteal (t-statistic = −4.157; *p* ≤ 0.01), (2) early follicular and mid-luteal (t-statistic = −4.257; *p* ≤ 0.01), (3) mid-follicular and mid-luteal (t-statistic = −2.992; *p* ≤ 0.05), (4) periovulatory and mid-luteal (t-statistic = −2.502; *p* ≤ 0.05), and (5) mid-luteal and late luteal (t-statistic = 3.704; *p* ≤ 0.05) (Table 2). These five differences, and no additional ones, were also detected as statistically significant in the realigned data pairwise comparisons (Table 2). Figure 3a shows the original, realigned, and imputed allopregnanolone trajectories across the menstrual cycle. Results of FDR-corrected pairwise comparisons for progesterone are presented in Appendix A, and visually depicted in Figure 3b.

### 3.4. Ratio of Allopregnanolone to Progesterone 

The most significant differences of the ratio were between (1) early follicular and mid-luteal (t-statistic = 23.709; *p* ≤ 0.001), (2) mid-follicular and mid-luteal (t-statistic = 19.535; *p* ≤ 0.001), followed by (3) early follicular and early luteal (t-statistic = 17.201; *p* ≤ 0.001), and (4) mid-follicular and early luteal (t-statistic = 15.967; *p* ≤ 0.001). The difference between early luteal and mid-luteal was statistically significant (t-statistic = 3.686, *p* ≤ 0.01). The trajectories of the original, realigned, and imputed data across the menstrual cycle for the ratio are represented in Figure 3c. Results of FDR-corrected pairwise comparisons for the ratio are presented in Table 3.

## 4. Discussion

The results of the present study show that allopregnanolone levels gradually increase from the start of the menstrual cycle through the mid-luteal subphase and dropping rapidly between the mid- and late luteal subphases. However, unlike its precursor progesterone, allopregnanolone mostly plateaus between the early and mid-luteal subphases as there are no significant differences in allopregnanolone levels between these two subphases. Nonetheless, mid-luteal allopregnanolone is still significantly higher relative to four other subphases, whereas early luteal allopregnanolone is only significantly higher than the early follicular allopregnanolone. Allopregnanolone:progesterone ratio is variable throughout the menstrual cycle, with the largest drop at the early luteal subphase, when progesterone levels rise rapidly. This likely reflects the notion that 5α-reductase already becomes saturated in the early luteal subphase. However, the enzymatic activity in the formation of allopregnanolone continues through the mid-luteal subphase, as allopregnanolone still forms, albeit at a far slower pace relative to progesterone. 

Allopregnanolone is considered a highly potent positive modulator of GABA_A_, enhancing both tonic and phasic GABA_A_–dependent inhibitory currents [29,30]. Hence, the neuroactive steroid hormone is associated with a number of neuropsychiatric, immune, and cardiometabolic conditions [13]. Female-limited epilepsy, for example, presents as a condition of low allopregnanolone levels, which is linked to seizure development [31]. The present study outlines the premenstrual trajectory of allopregnanolone in healthy reproductive-age women, which may serve as a future contrast of relevance to premenstrual conditions affected by circulating allopregnanolone. Perimenstrual catamenial epilepsy, the cyclical occurrence of seizure exacerbations near the time of menstruation, has been considered to be related to allopregnanolone withdrawal, which may impair GABAergic inhibition. The role of allopregnanolone in catamenial epilepsy, however, seems more complicated and not only dependent on circulating allopregnanolone levels. Herzog et al. [32] conducted a large-scale phase 3 clinical trial based on the premise that maintaining high allopregnanolone levels during the perimenstrual period would alleviate seizure frequency. However, in this trial, progesterone replacement therapy was not different than placebo [32]. A later study showed that the mid-luteal rise in progesterone, in fact, activates progesterone receptors and enhances excitatory neurotransmission, which would be counterbalanced in the menstrual cycle as long as high allopregnanolone levels maintain the GABAergic neurotransmission. Hence, progesterone treatment may have an anticonvulsant action through allopregnanolone, and a slower excitatory action that occurs following the activation of progesterone receptors and worsens seizures [33]. The present study clarifies the trajectory of allopregnanolone and the estimated 5-alpha reductase enzymatic activity involved in the formation of allopregnanolone across the menstrual cycle, which may be informative for numerous conditions, including different forms of epilepsy, associated with allopregnanolone levels, its biological effects, and the biological effects of 5-alpha reductase [13].

We identified significant differences in allopregnanolone concentrations between early follicular and early luteal, but four additional differences relative to the *mid*-luteal allopregnanolone (i.e., contrasted against early follicular, mid-follicular, periovulatory, and late luteal). We interpret this to mean that allopregnanolone starts to peak in the early luteal subphase but that the peak is reached in the mid-luteal subphase. However, this peak is proportionally much smaller than that of progesterone, which reaches its distinctive peak in the mid-luteal subphase, with the difference between the early and mid-luteal subphase highly significant (*p* ≤ 0.001) as shown in Appendix A. 

A previous study [34] examined the levels of allopregnanolone and progesterone on days 1–7, 13–16, and 20–23 in 10 premenopausal women. The authors selected these days to reflect the follicular phase, mid-cycle phase, and luteal phase. They showed that allopregnanolone levels significantly increased from the follicular to the luteal phase (*p* = 0.02). However, since luteinizing hormone was not measured, the boundary between periovulatory and luteal phase may have been misaligned. Moreover, the calendar method (i.e., definition of the luteal phase as days 20–23) may have reflected mid-, or late luteal subphase. Hence, the study [34] may have averaged out high allopregnanolone levels identified in the present study in the mid-luteal subphase, and the low levels of the late luteal subphase. 

The results of the present study must be interpreted with caution due to a small sample size of 30 participants with a confirmed ovulatory cycle, which had to be kept at a minimum given the current cost of the mass spectrometry analysis. However, a study sample size must be considered against possible sources of bias in the study population. In the present study, important sources of bias were removed as our study population, among other study criteria, was free of prescription and illicit drug intake, and did not smoke or consume large amounts of alcohol, which may distort circulating hormone levels across the cycle [35]. The strengths of our study include the implementation of the mass spectrometry methodology, which is the gold standard for steroid hormone measurement, and the realignment of study participants to the same biological window (i.e., subphase). Therefore, the present study generated a comprehensive dataset of high quality. Finally, we did not directly measure the activity of 5α-reductase. Such evaluation would require measurements of 5α-dihydroprogesterone. Instead, we relied on the literature demonstrating that the activity of 5α-reductase is a rate-limiting step in the conversion from progesterone to allopregnanolone [5,6]. Hence, our analysis approximates the activity of the rate-limiting enzyme 5α-reductase.

The current analysis improves the methodology of studying neuroactive steroid hormone trajectories across the menstrual cycle and outlines subphase differences in circulating allopregnanolone and allopregnanolone:progesterone ratio in reproductive-age women. This work advances our understanding of allopregnanolone trajectory and approximates the 5α-reductase activity across the entire menstrual cycle, which holds relevance in women’s health research.

## Figures and Tables

**Figure 1 biomolecules-13-00652-f001:**
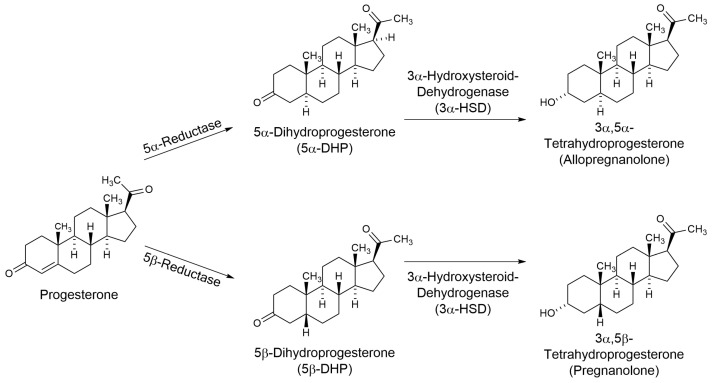
Biochemical pathways for the synthesis of allopregnanolone and pregnanolone.

**Figure 2 biomolecules-13-00652-f002:**
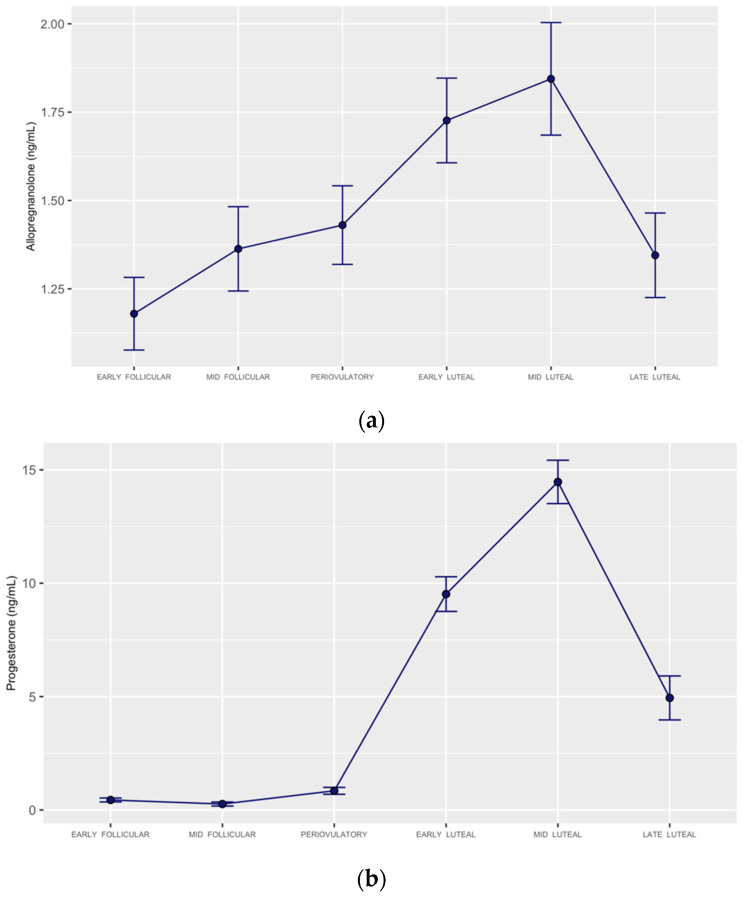
Allopregnanolone (**a**), progesterone (**b**), and allopregnanolone:progesterone ratio (**c**) mean and variance (SEM; presented as error bars) at each of the six subphases of the menstrual cycle.

**Figure 3 biomolecules-13-00652-f003:**
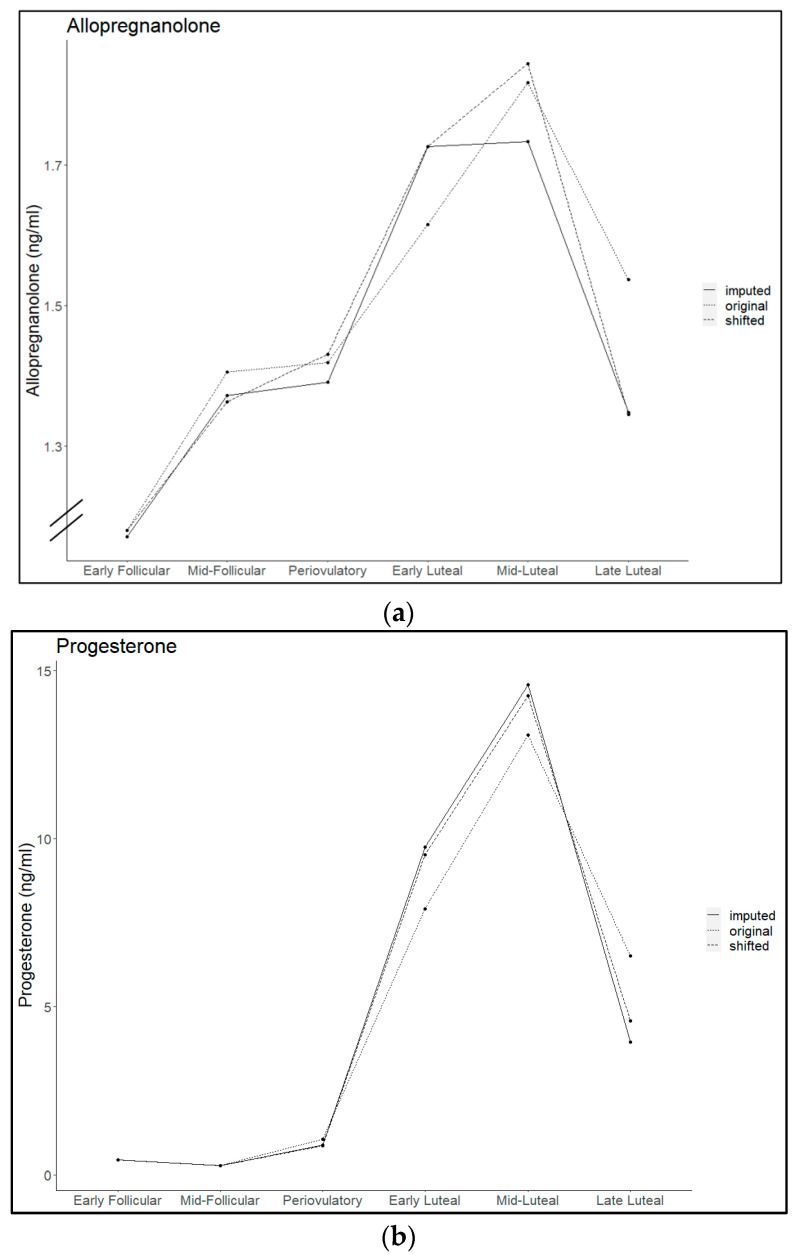
Circulating mean concentration changes for allopregnanolone (**a**), progesterone (**b**), and allopregnanolone: progesterone ratio (**c**) across the menstrual cycle from the original, shifted (realigned), and imputed datasets.

**Table 1 biomolecules-13-00652-t001:** Demographic and anthropomorphic characteristics of study participants with ovulatory cycle (*N* = 30).

Variable	Statistic
Age	26.40 (4.86)
Race	
White	12
Black or African American	6
American Indian/Alaska Native	1
Asian	7
Native Hawaiian or other Pacific Islander	0
More than one race	0
Unknown/do not want to specify	4
Ethnicity	
Hispanic	9
Non-Hispanic	19
Do not know/do not want to specify	2
Student Status	
Yes	16
No	14
Marital Status	
Single, never married	26
Married	3
Divorced	1
Income	
Less than USD 20,000	14
USD 20,000–34,999	5
USD 35,000–49,999	4
USD 50,000–74,999	4
>USD 74,999	3
Age of menarche	11.85 (1.48)
BMI	25.45 (4.80)

**Table 2 biomolecules-13-00652-t002:** Allopregnanolone subphase comparisons.

Timepoint Comparison		Class	t-Statistic	*p*-Value	Adjusted *p*-Value	Significance
Early follicular	Mid-follicular	Imputed	−5.210	2.15 × 10^−5^	3.22 × 10^−5^	***
Early follicular	Periovulatory	Imputed	1.028	0.3142	0.3142	ns
Early follicular	Early luteal	Imputed	17.201	1.61 × 10^−15^	8.05 × 10^−15^	***
Early follicular	Mid-luteal	Imputed	23.709	4.85 × 10^−18^	7.28 × 10^−17^	***
Early follicular	Late luteal	Imputed	6.045	3.88 × 10^−6^	7.28 × 10^−6^	***
Mid-follicular	Periovulatory	Imputed	3.574	0.0015	0.0016	**
Mid-follicular	Early luteal	Imputed	15.967	3.58 × 10^−15^	1.34 × 10^−14^	***
Mid-follicular	Mid-luteal	Imputed	19.535	1.12 × 10^−16^	8.42 × 10^−16^	***
Mid-follicular	Late luteal	Imputed	8.133	1.58 × 10^−8^	3.40 × 10^−8^	***
Periovulatory	Early luteal	Imputed	14.474	3.87 × 10^−11^	1.16 × 10^−10^	***
Periovulatory	Mid-luteal	Imputed	13.235	7.12 × 10^−11^	1.78 × 10^−10^	***
Periovulatory	Late luteal	Imputed	4.126	0.0005	0.0006	***
Early luteal	Mid-luteal	Imputed	3.686	0.0013	0.0015	**
Early luteal	Late luteal	Imputed	−4.128	0.0004	0.0005	***
Mid-luteal	Late luteal	Imputed	−5.995	7.28 × 10^−6^	1.21 × 10^−5^	***
Early follicular	Mid-follicular	Realigned	−4.580	0.0001	0.0002	***
Early follicular	Periovulatory	Realigned	1.509	0.1450	0.1450	ns
Early follicular	Early luteal	Realigned	22.650	9.75 × 10^−17^	7.31 × 10^−16^	***
Early follicular	Mid-luteal	Realigned	22.787	2.77 × 10^−17^	4.16 × 10^−16^	***
Early follicular	Late luteal	Realigned	5.254	9.72 × 10^−5^	0.0002	***
Mid-follicular	Periovulatory	Realigned	4.808	0.0001	0.0002	***
Mid-follicular	Early luteal	Realigned	21.454	8.86 × 10^−15^	4.43 × 10^−14^	***
Mid-follicular	Mid-luteal	Realigned	16.726	1.29 × 10^−13^	4.84 × 10^−13^	***
Mid-follicular	Late luteal	Realigned	8.090	1.20 × 10^−6^	2.57 × 10^−6^	***
Periovulatory	Early luteal	Realigned	14.575	4.08 × 10^−12^	1.13 × 10^−11^	***
Periovulatory	Mid-luteal	Realigned	13.421	4.51 × 10^−12^	1.13 × 10^−11^	***
Periovulatory	Late luteal	Realigned	2.743	0.0150	0.0171	*
Early luteal	Mid-luteal	Realigned	2.661	0.0160	0.0171	*
Early luteal	Late luteal	Realigned	−3.165	0.0080	0.0100	**
Mid-luteal	Late luteal	Realigned	−4.915	0.0002	0.0002	***

* *p* ≤ 0.05, ** *p* ≤ 0.01, *** *p* ≤ 0.001.

**Table 3 biomolecules-13-00652-t003:** Allopregnanolone: progesterone ratio subphase comparisons.

Timepoint Comparison		Class	t-Statistic	*p*-Value	Adjusted *p*-Value	Significance
Early follicular	Mid-follicular	Imputed	−5.210	2.15 × 10^−5^	3.22 × 10^−5^	***
Early follicular	Periovulatory	Imputed	1.028	0.3142	0.3142	ns
Early follicular	Early luteal	Imputed	17.201	1.61 × 10^−15^	8.05 × 10^−15^	***
Early follicular	Mid-luteal	Imputed	23.709	4.85 × 10^−18^	7.28 × 10^−17^	***
Early follicular	Late luteal	Imputed	6.045	3.88 × 10^−6^	7.28 × 10^−6^	***
Mid-follicular	Periovulatory	Imputed	3.574	0.0015	0.0016	**
Mid-follicular	Early luteal	Imputed	15.967	3.58 × 10^−15^	1.34 × 10^−14^	***
Mid-follicular	Mid-luteal	Imputed	19.535	1.12 × 10^−16^	8.42 × 10^−16^	***
Mid-follicular	Late luteal	Imputed	8.133	1.58 × 10^−8^	3.40 × 10^−8^	***
Periovulatory	Early luteal	Imputed	14.474	3.87 × 10^−11^	1.16 × 10^−10^	***
Periovulatory	Mid- luteal	Imputed	13.235	7.12 × 10^−11^	1.78 × 10^−10^	***
Periovulatory	Late luteal	Imputed	4.126	0.0005	0.0006	***
Early luteal	Mid- luteal	Imputed	3.686	0.0013	0.0015	**
Early luteal	Late luteal	Imputed	−4.128	0.0004	0.0005	***
Mid-luteal	Late luteal	Imputed	−5.995	7.28 × 10^−6^	1.21 × 10^−5^	***
Early follicular	Mid-follicular	Realigned	−4.580	0.0001	0.0002	***
Early follicular	Periovulatory	Realigned	1.509	0.1450	0.1450	ns
Early follicular	Early luteal	Realigned	22.650	9.75 × 10^−17^	7.31 × 10^−16^	***
Early follicular	Mid-luteal	Realigned	22.787	2.77 × 10^−17^	4.16 × 10^−16^	***
Early follicular	Late luteal	Realigned	5.254	9.72 × 10^−5^	0.0002	***
Mid-follicular	Periovulatory	Realigned	4.808	0.0001	0.0002	***
Mid-follicular	Early luteal	Realigned	21.454	8.86 × 10^−15^	4.43 × 10^−14^	***
Mid-follicular	Mid-luteal	Realigned	16.726	1.29 × 10^−13^	4.84 × 10^−13^	***
Mid-follicular	Late luteal	Realigned	8.090	1.20 × 10^−6^	2.57 × 10^−6^	***
Periovulatory	Early luteal	Realigned	14.575	4.08 × 10^−12^	1.13 × 10^−11^	***
Periovulatory	Mid-luteal	Realigned	13.421	4.51 × 10^−12^	1.13 × 10^−11^	***
Periovulatory	Late luteal	Realigned	2.743	0.0150	0.0171	*
Early luteal	Mid-luteal	Realigned	2.661	0.0160	0.0171	*
Early luteal	Late luteal	Realigned	−3.165	0.0080	0.0100	**
Mid-luteal	Late luteal	Realigned	−4.915	0.0002	0.0002	***

* *p* ≤ 0.05, ** *p* ≤ 0.01, *** *p* ≤ 0.001.

## Data Availability

The data presented in this study are available by request from the corresponding author. The data are not publicly available to preserve scientific integrity of research methodology.

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
