# Peer review of "Trajectories of Allopregnanolone and Allopregnanolone to Progesterone Ratio across the Six Subphases of Menstrual Cycle"

_biomolecules, 2023, doi:10.3390/biom13040652_

Round 1
Reviewer 1 Report
This is a beautifully conducted study in which sophisticated methods were used to align individual menstrual cycles with menstrual cycle base and both allopregnanolone and progesterone were measured by tandem mass spectrometry. Using these methods, they were able to characterize allopregnanolone and the allopregnanolone/progesterone ratio throughout the menstrual cycle. The results are interesting and an important addition to the literature. It is a strength of the study that the original, imputed and shifted data are all presented. A few comments follows:
1. The legend of Figure 3 states that progesterone levels and the allopregnanolone:progesterone ratio across menstrual cycle are shown in addition to the allopregnanolone levels. However, the progesterone levels and the allopregnanolone:progesterone ratio are missing. Please add these important data to this figure, as was intended. Presumably mean (not median) levels are reported in this figure? Please indicate.
2. In Figures 2 and 3, the Y axis does not cross zero, which can be indicated by slashes through the Y axis or whatever the convention is for this journal.
3. In Figure 2, please indicate what the error bars are delineating (SEM?) I recommend including progesterone and allopregnanolone: progesterone ratios in this figure also (in separate panels).
Author Response
Reviewer 1
COMMENT: This is a beautifully conducted study in which sophisticated methods were used to align individual menstrual cycles with menstrual cycle base and both allopregnanolone and progesterone were measured by tandem mass spectrometry. Using these methods, they were able to characterize allopregnanolone and the allopregnanolone/progesterone ratio throughout the menstrual cycle. The results are interesting and an important addition to the literature. It is a strength of the study that the original, imputed and shifted data are all presented. A few comments follows:
RESPONSE: We thank the reviewer for taking the time to provide his/her thoughtful comments. Please see our responses below.
COMMENT #1: The legend of Figure 3 states that progesterone levels and the allopregnanolone:progesterone ratio across menstrual cycle are shown in addition to the allopregnanolone levels. However, the progesterone levels and the allopregnanolone:progesterone ratio are missing. Please add these important data to this figure, as was intended. Presumably mean (not median) levels are reported in this figure? Please indicate.
RESPONSE #1: Thank you for your comment. We apologize that the two figures were left out in the copy/paste process to the publisher template. They are included in the new version, along with the specification that the mean levels are presented.
COMMENT #2: In Figures 2 and 3, the Y axis does not cross zero, which can be indicated by slashes through the Y axis or whatever the convention is for this journal.
RESPONSE #2: We have added the slashes to the figures where the Y axis does not cross 0.
COMMENT #3: In Figure 2, please indicate what the error bars are delineating (SEM?) I recommend including progesterone and allopregnanolone: progesterone ratios in this figure also (in separate panels).
RESPONSE #3: In response to the reviewer’s comments, we have revised Figure 2 to now include Figure 2a (allopregnanolone), 2b (progesterone), and 2c (allopregnanolone:progesterone) ratio. In addition, it is clarified that the error bars represent the SEM.
Reviewer 2 Report
The study by Hamidovic and colleagues characterizes the allopregnanolone:progesterone ratio across six phases of the menstrual cycle in women. This time course of steroid levels across the menstrual cycle add some novel insights to our understanding of the physiological role of allopregnanolone in ovarian function, however, is mostly confirmatory of a previous study by another group that differentiates between the early- and mid-follicular phase, and among the early-, mid- and late-luteal phase.
There are several concerns as follow:
-the aim of the study at the end of the introduction (lines 89-90):” to clarify the changes of allopregnanolone and the enzymatic activity involved in its synthesis across the entire menstrual cycle” is not accomplished. No enzymatic studies are presented,
-the allopregnanolone:progesterone ratio cannot be used as a measure for enzymatic activity; two different enzymes are involved in the conversion of progesterone to allopregnanolone, therefore the conclusion referring to the activity of 5a-reductase across menstrual cycle is highly speculative. In my opinion this story needs further work.
-the discussion relative to the putative role of allopregnanolone in catamenial epilepsy (lines 318-328) is not clear . What’s the point they want to make? Once again the conclusion of this paragraph is not supported by data.
Author Response
Reviewer 2
COMMENT: The study by Hamidovic and colleagues characterizes the allopregnanolone:progesterone ratio across six phases of the menstrual cycle in women. This time course of steroid levels across the menstrual cycle add some novel insights to our understanding of the physiological role of allopregnanolone in ovarian function, however, is mostly confirmatory of a previous study by another group that differentiates between the early- and mid-follicular phase, and among the early-, mid- and late-luteal phase.
RESPONSE: We thank the reviewer for taking the time to provide his/her thoughtful comments. Please see our responses below.
There are several concerns as follow:
COMMENT #1: the aim of the study at the end of the introduction (lines 89-90):” to clarify the changes of allopregnanolone and the enzymatic activity involved in its synthesis across the entire menstrual cycle” is not accomplished. No enzymatic studies are presented,
RESPONSE #1: We thank the reviewer for this comment and clarify that the study’s meaning of enzymatic activity involves calculating the ratio of one steroid hormone to another (i.e., allopregnanolone:progesterone).
COMMENT #2: the allopregnanolone:progesterone ratio cannot be used as a measure for enzymatic activity; two different enzymes are involved in the conversion of progesterone to allopregnanolone, therefore the conclusion referring to the activity of 5a-reductase across menstrual cycle is highly speculative. In my opinion this story needs further work.
RESPONSE #2: We agree with the reviewer that this area of research needs further work. We have revised the following paragraph (lines 47-56) to emphasize the word “estimated” and clarify our approach:
“The activity of 5α-reductase is considered a rate-limiting step in the conversion from progesterone to allopregnanolone [4, 5], based on the results of immunohistochemical studies showing that the allopregnanolone decrease in corticolimbic areas [6-8] is related to a decreased expression of 5α-reductase [7, 9, 10]. Whether, and when, this occurs in the menstrual cycle is not clear, and, importantly, whether the estimated activity of 5α-reductase ceases at any point of the menstrual cycle is not known. The activity of 5α-reductase is estimated in the present study since it is not directly involved in the formation of allopregnanolone from progesterone (Figure 1).”
COMMENT #3: the discussion relative to the putative role of allopregnanolone in catamenial epilepsy (lines 318-328) is not clear. What’s the point they want to make? Once again the conclusion of this paragraph is not supported by data.
RESPONSE #3: In response to the reviewer’s comment, we have revised this paragraph for clarity, which now reads:
“Allopregnanolone is considered a highly potent positive modulator of GABAA, enhancing both tonic and phasic GABAA – dependent inhibitory currents [28, 29]. Hence, allopregnanolone is associated with a number of neuropsychiatric, immune, and cardiometabolic conditions [12]. Female-limited epilepsy, for example, presents as a condition of low allopregnanolone levels, which is linked to seizure development [30]. Hence, the present study outlines the premenstrual trajectory of allopregnanolone in healthy reproductive age women, which may serve as a future contrast of relevance to premenstrual conditions affected by circulating allopregnanolone. Perimenstrual catamenial epilepsy, the cyclical occurrence of seizure exacerbations near the time of menstruation, has been considered to be related to allopregnanolone withdrawal, which would impair GABAergic inhibition. The role of allopregnanolone in catamenial epilepsy, however, seems more complicated and not only dependent on circulating allopregnanolone levels. Herzog et al [31] conducted a large-scale phase 3 clinical trial based on the premise that maintaining high allopregnanolone levels during the perimenstrual period would alleviate seizure frequency. However, in this trial, progesterone replacement therapy was not different than placebo [31]. A later study showed that the mid-luteal rise in progesterone, in fact, activates progesterone receptors and enhances excitatory neurotransmission, which would be counterbalanced in the menstrual cycle as long as high allopregnanolone levels maintain the GABAergic neurotransmission. Hence, progesterone treatment may have an anticonvulsant action through allopregnanolone, and a slower excitatory action that occurs following the activation of progesterone receptors and worsens seizures [32]. The present study clarifies the trajectory of allopregnanolone and the estimated 5-alpha reductase enzymatic activity involved in the formation of allopregnanolone across the menstrual cycle, which may be informative for numerous conditions, including different forms of epilepsy, associated with allopregnanolone levels, its biological effects, or the biological effects of 5-alpha reductase [12].”
Reviewer 3 Report
This interesting manuscript provides the first comprehensive overview of allopregnanolone levels across the detailed 6 subphases of the menstrual cycle in humans. As a result it is a contribution to the literature and can serve as a standard against which other menstrual cycle studies can compare results.
I do have some suggestions for revision.
1. In the introduction, it would be helpful to explain a bit more about the conversion process from P4 to ALLO. The figure implies that the action of 3aHSD can convert BOTH 5aDHP AND 5bDHP to either ALLO or pregnanolone. What determines, then, which molecule is formed by 3aHSD? You also make no mention of pregnanolone, which is confusing.
2. There is no information in the methods about a power calculation, so it is hard to judge whether this small sample size is sufficient for the purposes for which it is used. I realize you are using within-subject comparisons, but since you do lump all subjects data together to come up with mean levels across time, it would be helpful to know how many participants would be needed to detect these differences.
3. There is also no information about how you have statistically handled the issue of multiple comparisons.
4. It would be helpful to provide the ALLO:P4 ratio across time in a figure, as you have done for ALLO in Figure 2 -- the tables are long and somewhat overwhelming and a figure will be much more clear.
5. I think it is important to draw back on the language (throughout the manuscript) claiming that this study has allowed you to determine something about 5a reductase activity. It has not -- you've shown how ALLO rises and falls and how that is related to the rise and fall of progesterone, but we have no information about whether this implies 5a reductase activity as opposed to 3aHSD. As you point out in the intro, the data showing that the step catalyzed by 5a reductase is the rate-limiting one comes from animal literature, and we really don't know if that applies here.
Author Response
Reviewer 3
COMMENT: This interesting manuscript provides the first comprehensive overview of allopregnanolone levels across the detailed 6 subphases of the menstrual cycle in humans. As a result it is a contribution to the literature and can serve as a standard against which other menstrual cycle studies can compare results.
RESPONSE: We thank the reviewer for taking the time to provide his/her thoughtful comments. Please see our responses below.
I do have some suggestions for revision.
COMMENT #1: In the introduction, it would be helpful to explain a bit more about the conversion process from P4 to ALLO. The figure implies that the action of 3aHSD can convert BOTH 5aDHP AND 5bDHP to either ALLO or pregnanolone. What determines, then, which molecule is formed by 3aHSD? You also make no mention of pregnanolone, which is confusing.
RESPONSE #1: In response to the reviewer’s comment, we have regraphed Figure 1 to clearly show that 3α-hydroxysteroid dehydrogenase (3α-HSD) converts 5α-DHP and 5β-DHP to allopregnanolone and pregnanolone, respectively. We have referenced pregnanolone, both in the manuscript text, as well as in the figure description. We also added the following (last sentence in the paragraph):
“Allopregnanolone is synthesized from progesterone in two steps (Fig 1). First, 5α- and 5β-reductases convert progesterone to 5α-dihydroprogesterone (5α-DHP) and 5β-dihydroprogesterone (5β-DHP), respectively, following which 3α-hydroxysteroid dehydrogenase (3α-HSD) converts 5α-DHP and 5β-DHP to allopregnanolone and pregnanolone, respectively [2, 3]. Which pregnanedione (5α- or 5β-) is a substrate for 5α-HSD has not been specifically examined to date, but the process is presumably related to the differential tissue distribition of 5α-DHP or 5β-DHP, which is, in turn, determined by differential expression profiles of the steroidogenic enzymes 5α- and 5β-reductases [4].”
COMMENT #2: There is no information in the methods about a power calculation, so it is hard to judge whether this small sample size is sufficient for the purposes for which it is used. I realize you are using within-subject comparisons, but since you do lump all subjects data together to come up with mean levels across time, it would be helpful to know how many participants would be needed to detect these differences.
RESPONSE #2: We added the following the the revised manuscipt:
“We conducted a power analysis to test the hypothesis for one independent group mean against the null hypothesis using a two-tailed test, a medium effect size (d = .50) [35], and an alpha of .05. Results showed that a total sample of 32 participants was required to achieve a power of 0.80.”
COMMENT #3: There is also no information about how you have statistically handled the issue of multiple comparisons.
RESPONSE #3: Multiple comparisons were handled by the use of the false discovery rate. Line 259:
“We conducted pairwise comparisons of all timepoints for both the realigned and imputed data, and we used False Discovery Rate (FDR) to correct for multiple comparisons.”
COMMENT #4: It would be helpful to provide the ALLO:P4 ratio across time in a figure, as you have done for ALLO in Figure 2 -- the tables are long and somewhat overwhelming and a figure will be much more clear.
RESPONSE #4: We have provided figures of the ratio as Figures 2 (bottom) and 3 (bottom).
COMMENT #5: I think it is important to draw back on the language (throughout the manuscript) claiming that this study has allowed you to determine something about 5a reductase activity. It has not -- you've shown how ALLO rises and falls and how that is related to the rise and fall of progesterone, but we have no information about whether this implies 5a reductase activity as opposed to 3aHSD. As you point out in the intro, the data showing that the step catalyzed by 5a reductase is the rate-limiting one comes from animal literature, and we really don't know if that applies here.
RESPONSE #5: We thank the reviewer for this comment. In response, we have revised this point in serveral parts of the manuscript.
Instance 1 (Abstract, line 34): Hence, the estimated activity of 5α‐reductase decreases, but does not cease, at any point across the menstrual cycle.
Instance 2 (Introduction, line 57): Whether, and when, this occurs in the menstrual cycle is not clear, and, importantly, whether the estimated activity of 5α-reductase ceases at any point of the menstrual cycle is not known. The activity of 5α-reductase is estimated in the present study since it is not directly involved in the formation of allopregnanolone from progesterone (Figure 1).”
Instance 3: (Discussion, line 399): The present study clarifies the trajectory of allopregnanolone and the estimated 5-alpha reductase enzymatic activity involved in the formation of allopregnanolone across the menstrual cycle, which may be informative for numerous conditions, including different forms of epilepsy, associated with allopregnanolone levels, its biological effects, or the biological effects of 5-alpha reductase [13].
Present in the original manuscript
Limitation: Finally, we did not directly measure the activity of 5α-reductase. Such evaluation would require measurements of 5α-dihydroprogesterone. Instead, we relied on the literature demonstrating that the activity of 5α-reductase is a rate-limiting step in the conversion from progesterone to allopregnanolone [5, 6]. Hence, our analysis approximates the activity of the rate-limiting enzyme 5α-reductase.
Conclusion: The current analysis improves the methodology of studying neuroactive steroid hormone trajectories across the menstrual cycle and outlines subphase differences in circulating allopregnanolone and allopregnanolone:progesterone ratio in reproductive age women. This work advances our understanding of allopregnanolone trajectory and approximates the 5α-reductase activity across the entire menstrual cycle, which holds relevance in women’s health research.
Round 2
Reviewer 2 Report
The paper can be accepted in the revised form
Author Response
Thank you for the opportunity to revise the overlapping methods from our earlier work. We have now revised several sections to prevent the overlap.